# Predicting age of respiratory syncytial virus infection from birth timing

Chris G. McKennan [1] ✉, Tebeb Gebretsadik[2], Steven M. Brunwasser[3], Michael Nodzenski[4], Daniel J. Jackson[5], James E. Gern [5], Pingsheng Wu [2,6] & Tina V. Hartert [6,7] ✉

Respiratory syncytial virus (RSV) infects nearly all children by age 2 to 3 years, and early-life infection—defined using active and passive surveillance with quantitative polymerase chain reaction- and serology-identified infection—has been implicated as a causal factor in childhood asthma. As such, identifying infants that are likely to be infected with RSV during this critical susceptibility window has important implications for identifying individuals at risk for chronic respiratory sequelae. However, determining the age of RSV infection in large populations is challenging because many infections are asymptomatic, making accurate detection dependent on intensive and costly surveillance. To address this, we developed a probability model for age of first RSV infection. It uses an infant's birthdate, demographic covariates, and publicly available RSV circulation data to determine the probability they were first infected at any age from birth to one year. Our model is interpretable, accounts for nearly 37% of the variance in age at first infection, and generalizes across four independent datasets collected from participants in the United States, where we use it to accurately predict age of first infection in two independent cohorts. Our work facilitates reliable estimation of the age of infant RSV infection during the first year of life without the need for active surveillance.

Respiratory syncytial virus (RSV) is a leading cause of respiratory illness in infants and young children, with about half of all infants infected by age one year and nearly all by age three[1–3]. The timing of first infection is critical—not only for acute disease severity but also for long-term respiratory health. Early-life RSV infection has been linked to increased risk of childhood respiratory disease. We previously showed that infants infected with RSV before age one—identified through active and passive surveillance using quantitative polymerase chain reaction and serology—are significantly more likely to develop asthma by age five[3], and that this association may be mediated by the effects of RSV on airway barrier development, metabolism, the microbiome, and epigenetics[4–8]. In addition, we found that the date of birth relative to RSV circulation is associated with asthma risk, with the highest incidence among children born roughly four months before the seasonal RSV peak[9]. These findings remain relevant even in the era of RSV prevention products: although RSV vaccines and RSV monoclonal antibodies reduce disease severity[10,11], they do not appear to prevent infection[12].

Together, these findings underscore the critical need to identify the age of infant RSV infection. Doing so could advance epidemiological research by clarifying RSV's causal role in respiratory outcomes, immune and airway development, and long-term susceptibility to

[1]Department of Statistics, University of Pittsburgh; Pittsburgh, Pennsylvania, USA. [2]Department of Biostatistics, Vanderbilt University Medical Center; Nashville, Tennessee, USA. [3]Department of Psychology, Rowan University; Glassboro, New Jersey, USA. [4]Rho Inc, Federal Systems Division, Durham, North Caroline, USA. [5]Department of Pediatrics, University of Wisconsin School of Medicine and Public Health, Madison, Wisconsin, USA. [6]Department of Medicine, Vanderbilt University Medical Center; Nashville, Tennessee, USA. [7]Department of Pediatrics, Vanderbilt University Medical Center; Nashville, Tennessee, USA. ✉e-mail: chm195@pitt.edu; tina.hartert@vumc.org

chronic disease—particularly by pinpointing when RSV infection has the greatest impact. This has important public health implications, informing prevention strategies aimed at reducing infection during high-risk periods. However, determining the age of first infection in large cohorts is challenging due to the high rate of asymptomatic cases, the likelihood of infections going undetected, and the tremendous cost and complexity of active surveillance[3,13].

As an alternative to costly surveillance, accurate in silico prediction of age at first RSV infection could enable retrospective and prospective analyses of RSV-asthma relationships in existing and future cohorts. While models that predict severe RSV bronchiolitis requiring healthcare visits do exist[9,14,15], they do not capture the full burden of RSV infection. Most infections are mild or asymptomatic and do not result in medical encounters[3,16], making these models poorly suited for estimating general infection risk.

To address this gap, we developed a probability model to estimate the age of first RSV infection in infancy. Our model leverages a child's birthdate, basic demographic data, and publicly available RSV circulation data from the United States Centers for Disease Control and Prevention (CDC) to estimate the probability of first infection at each week of age from birth to one year. The model is based on the well-characterized seasonal pattern of RSV, which typically peaks in winter and subsides in summer in temperate climates. For instance, an infant born in July is likely to reach six months of age during January, when RSV circulation is high-resulting in a high probability of first infection at that time.

In this study, we applied and validated our model across four independent cohorts from the United States using RSV surveillance data. Specifically, we evaluated the model's fit to observed data, its ability to predict the age of first RSV infection, and its potential utility as a scalable alternative to active surveillance.

## Results

### Cohort demographics and RSV abundance over time

Table 1 provides an overview of the study designs and demographic information for the four cohorts considered in this study. Birthdates were approximately uniform across the birth months considered in each cohort (Supplementary Fig. S1). Additional details regarding the cohorts used in this study have been previously reported[17–20].

We used RSV circulation data collected by the CDC to determine the abundance of RSV from January 1995 to April 2020 for all geographical regions in the United States (see Methods). Figure 1A contains an example of the raw circulation data for the region and time period covering INSPIRE participants. Figure 1B is the output of our smoothing and standardization pipeline, where the abundance of RSV is assumed to be proportional to the y-axis in Fig. 1B.

### Parameter estimates and model evaluation

Our model is based off a model for the instantaneous risk of first RSV infection, also called the "hazard function" in survival analysis. It assumes an infant's risk of first being infected at age $a$ equals the known abundance of RSV at the time they are age $a$ (which depends on their birthdate) times an unknown age-dependent weight function $w(a)$. This function captures age-related variation in RSV infection risk at a fixed point in time. Notably, $w(a)$ will be small if infants are not at risk of being infected at age $a$ and large if they are at risk. Non-birthdate covariates are incorporated using a proportional hazards assumption (see Methods).

We used the INSPIRE cohort to estimate model parameters. Figure 2A shows that the weight function is small for ages close to zero, indicating newborns are less likely to be infected by RSV than older infants. The function peaks at around 6.5 months, suggesting this may be the time infants are at the greatest risk of infection. While it appears to decrease after eight months, the wide confidence intervals and dearth of infections during these months suggest the decrease during

this time cannot be confirmed. A sensitivity analysis indicated the shape of the weight function, e.g., its maxima at 2 months and 6.5 months and nadir at 4 months, was consistent when we re-estimated the weight function after partitioning INSPIRE individuals by race, sex, daycare attendance, or number of older siblings (Section S1.2). Figure 2B gives the estimate for the effect of non-birthdate covariates expressed as a hazard ratio, where three of the four are clearly important in predicting age of first infection. Having estimated parameters, we then investigated the relationship between birthdate and infection risk, as well as how the relationship changed with non-birthdate covariates. As indicated in Fig. 2C, infants born in June (summer) are most likely to be infected at 6 months, whereas those born in December (late fall, early winter) are most likely to be first infected at 1–3 months. Non-birthdate covariates appear to only alter the magnitude, not the shape, of the relationship between birthdate and infection risk.

We next sought to determine the relative importance of birthdate and the non-birthdate covariates. As Fig. 2D indicates, the importance depends on the outcome one is trying to predict. If the goal is to try to predict whether an infant is infected by age one year, then birthdate and non-birthdate covariates have similar modest impacts. However, if it is known that an infant is infected by age one year and the goal is to predict the exact age of first infection, birthdate is unequivocally more important and explains nearly 37% of the variance in infection age. The impact of birthdate on the outcome "infection by age one year" is dramatically smaller because all infants, regardless of birthdate, experience at least one RSV season during a one-year interval. Section S2.7 details how we computed the percent variance explained.

We next evaluated how well our model fit the observed data in INSPIRE. Figure 2E gives the estimated probability density functions for the age of first infection for individuals born in June through December. By design, no INSPIRE individuals were born in January through May. These estimates are strikingly accurate, which is quite remarkable given that the effective number of parameters used to fit them was only 10.6 (6.6 parameters from $w(a)$ and 4 parameters from non-birthdate covariates)[21].

### Testing our estimated model in two independent cohorts

To evaluate our model's prediction accuracy, we used our estimates derived from INSPIRE to predict age of first infection in the COAST and URECA cohorts. Specifically, we sought to predict (i) which individuals were infected by age one year, as measured by RSV surveillance and serology, and (ii) the exact age of first infection for individuals with known age of first infection. For (i), we were particularly interested in assessing whether our model was well-calibrated. For example, if our model reports an individual was infected by age one year with probability 0.3, then 30% of all individuals with that infection probability should actually be infected by year one. Figure 3A gives the calibration plot and suggests our model is well-calibrated, as points lie on or near the dashed red line and 18 out of 20 points (90%) lie within the 95% confidence intervals (dashed gray lines). We combine COAST and URECA in Fig. 3A to ensure probability bins had a sufficient number of individuals. Figure S4 gives separate calibration plots for COAST and URECA individuals.

For (ii), we evaluated whether our probability density function estimate for age of first infection mirrored the distribution of observed first infection ages in the COAST cohort. URECA individuals did not have the age of first infection data. We grouped COAST individuals by birth month, where we defined the "risk month" and "non-risk month" groups to be those born in June through December and January through May, respectively. We could not consider finer partitions of birth months because of sample size limitations in COAST. Figure 3B gives the probability density estimates in these two groups, which closely resemble the observed distributions.

**Table 1 | Study designs and demographic information for the four cohorts in our study**

| | INSPIRE (n = 1741) | COAST (n = 242) | URECA (n = 301) | PRIMA (n = 152, 622) |
|---|---|---|---|---|
| **Study design** | | | | |
| Birth months | June - December | January - December | January - December | January - December |
| Study period (First birth - last birth 1 year) | June 2012 - December 2014 | November 1998 - May 2001 | February 2005 - March 2008 | January 1995 - December 2004 |
| Study location (%) | Tennessee (100%) | Wisconsin (100%) | Baltimore, Maryland (28%) | Tennessee (100%) |
| | | | New York, New York (13%) | |
| | | | Boston, Massachusetts (28%) | |
| | | | St. Louis, Missouri (31%) | |
| RSV serology at one year of age? | Yes | Yes | Yes | No |
| Surveillance during first year of life | RSV qPCR following minimal infection symptoms | RSV qPCR following minimal infection symptoms | N/A | Bronchiolitis healthcare encounters |
| Number of individuals with known age of first RSV infection (%) | 361 (21%) | 53 (22%) | 0 (0%) | 0 (0%) |
| **Demographic data** | | | | |
| **Race/ethnicity: n (%)** | | | | |
| Black | 308 (18%) | 7 (3%) | 224 (74%) | 55,862 (37%) |
| Hispanic | 156 (9%) | 0 (0%) | 52 (17%) | 6,374 (4%) |
| White | 1131 (65%) | 233 (96%) | 2 (1%) | 84,053 (55%) |
| Other/Missing | 146 (8%) | 2 (1%) | 23 (8%) | 6,333 (4%) |
| **Sex: n (%)** | | | | |
| Female | 828 (48%) | 103 (43%) | 151 (50%) | 73,930 (48%) |
| Male | 913 (52%) | 139 (57%) | 150 (50%) | 78,692 (52%) |
| **Daycare attendance in first year of life: n (%)** | | | | |
| Yes | 578 (33%) | 128 (53%) | 98 (33%) | 0 (0%) |
| No | 1135 (65%) | 114 (47%) | 203 (67%) | 0 (0%) |
| Missing | 28 (2%) | 0 (0%) | 0 (0%) | 152,622 (100%) |
| **Older siblings: n (%)** | | | | |
| Yes | 876 (50%) | 132 (55%) | 67 (22%) | 90,136 (59%) |
| No | 865 (50%) | 110 (45%) | 232 (77%) | 62,373 (41%) |
| Missing | 0 (0%)) | 0 (0%)) | 2 (1%)) | 113 (< 0.1%)) |
| **Maternal asthma: n (%)** | | | | |
| Yes | 342 (20%) | 100 (41%) | 135 (45%) | 3,884 (2.5%) |
| No | 1398 (80%) | 142 (59%) | 165 (55%) | 83,264 (55%) |
| Missing | 0 (0%) | 0 (0%) | 1 (0.3%) | 65,474 (43%) |
| **Breastfeeding: n (%)** | | | | |
| Ever breastfed | 1383 (79%) | 196 (81%) | 154 (51%) | 0 (0%) |
| Never breastfed | 345 (20%) | 46 (19%) | 142 (47%) | 0 (0%) |
| Missing | 13 (1%) | 0 (0%) | 5 (2%) | 152,622 (100%) |
| **Insurance: n (%)** | | | | |
| Private | 795 (46%) | 0 (0%) | 16 (5%) | 0 (0%) |
| Public | 923 (53%) | 0 (0%) | 277 (92%) | 152,622 (100%) |
| Other/Missing | 23 (1%) | 242 (100%) | 8 (3%) | 0 (0%) |

The total number of individuals in INSPIRE, COAST, and URECA is the number of individuals known to be infected or not infected with RSV by age one year. The total in PRIMA is the number of children with an asthma diagnosis at age six years.

## Using bronchiolitis healthcare visits to estimate model parameters

As there are few RSV surveillance studies, we lastly sought to determine whether bronchiolitis healthcare visits from the PRIMA cohort could be used to estimate model parameters. Due to the availability of birth and healthcare visit timing data, the weight function $w(a)$ was estimated assuming it was a step function with steps (i.e., discontinuities) at each month (see "Methods"). We did not consider non-birthdate covariates because we found their impact in INSPIRE to be

minor. Notably, these data required estimating a nuisance function $c(a)$, which is the probability an infection at age $a$ becomes bronchiolitis requiring a healthcare visit (see "Methods"). This helped avoid biasing our estimate for $w(a)$, as the age distributions of general RSV infections and infections required a healthcare visit are likely different[22].

Figure 4 A plots the estimated weight function. While it is similar to the estimate from INSPIRE, the uncertainty in the estimate from PRIMA is substantially larger, where one would need to increase the

A)

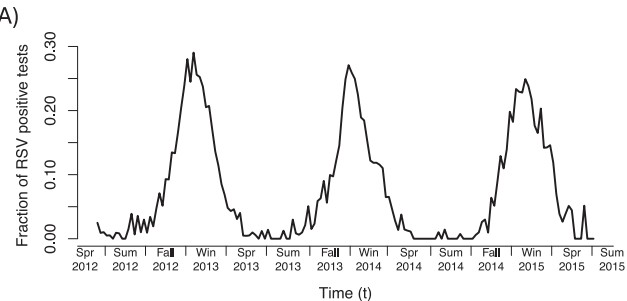
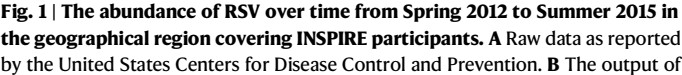

B)

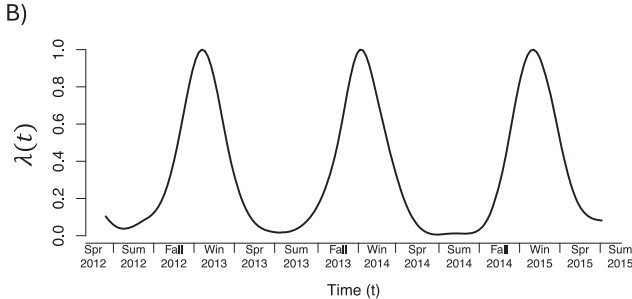

**Fig. 1 | The abundance of RSV over time from Spring 2012 to Summer 2015 in the geographical region covering INSPIRE participants. A** Raw data as reported by the United States Centers for Disease Control and Prevention. **B** The output of our smoothing and standardization pipeline applied to the raw data in A). The y-axis, $\lambda(t)$, is assumed to be proportional to be the amount of circulating RSV at time $t$ in a given geographical region.

number of PRIMA study individuals from $1.5 \times 10^5$ to $1.5 \times 10^6$ to obtain standard errors equal to those from INSPIRE (see Section S1.3 for details). This considerable uncertainty stems from the fact that when using bronchiolitis healthcare visits, one must additionally estimate the nuisance function $c(a)$. As $c(a)$ is highly non-uniform (see Section S1.3), it is difficult to distinguish between healthcare visits that are due to an increase in infection risk (i.e., a change in $w(a)$) from those arising because individuals were more prone to bronchiolitis (i.e., a change in $c(a)$). Consequently, estimates for $w(a)$ and $c(a)$ were highly correlated.

Despite the increase in uncertainty, we evaluated whether our estimate for $w(a)$ from PRIMA could be used to model the age of first RSV infection in INSPIRE individuals. As PRIMA and INSPIRE are independent cohorts, this evaluation was equivalent to testing our model in an independent cohort. Figure 4B shows that model-based probabilities using the PRIMA-derived $w(a)$ (black curves) closely resemble first infection frequencies observed in INSPIRE (blue curves). These probabilities were just as accurate as those obtained using the INSPIRE-derived $w(a)$ (Fig. 4C), which is quite remarkable given that INSPIRE-derived probabilities and observed frequencies were obtained from the same individuals.

## Discussion

In this study, we propose a probability model that utilizes publicly available RSV circulation data from the CDC to model the age of first RSV infection as a function of birthdate and other demographic covariates. Our approach was to model the instantaneous risk of first infection, also known as the "hazard function" in the field of survival analysis. It takes an infant's risk of being first infected at age $a$ to be the product of the known abundance of RSV when they are age $a$, which is determined by their birthdate and RSV surveillance data from the CDC, and an unknown age-dependent weight function. The weight function captures the age-dependent variation in RSV infection risk at a fixed point in time. Since we model hazard functions, we used the proportional hazards assumption to incorporate non-birthdate covariates. We could not use a standard Cox proportional hazard analysis to predict infection age because the proportional hazard assumption did not hold for the birthdate covariate.

We took a non-parametric approach to model the weight function and estimated it—along with the effects of non-birthdate covariates—using RSV surveillance data from the INSPIRE cohort. The resulting fitted model accurately captured the relationship between birthdate and age at first infection despite the model only having 10.6 effective parameters[21]. Our estimates for the impact of non-birthdate covariates were consistent with those from a recent study examining risk factors for RSV infection before age one, based on serological testing[2].

We tested our estimated model in two independent cohorts, demonstrating that the model-derived probabilities of infection by age one year were well-calibrated. In addition, the estimated distributions

of infection ages for infants known to be infected by age one closely mirrored the observed distributions. Notably, the two validation cohorts differed substantially from INSPIRE in racial composition, geographic location, study period, and birth months (Table 1), supporting the generalizability of our model, its parameters (e.g., the weight function), and resulting estimates across diverse populations.

The generalizability of our model and its ability to quantify the predictive value of birthdate and non-birthdate covariates (Fig. 2D) suggest a way to conduct early-life "RSV surveillance" to capture age of infection without active monitoring. Notably, for infants known to be infected in the first year, birthdate alone explains nearly 37% of the variance in age of infection. A feasible study design could therefore involve serological testing at age one year to determine prior infection status, then applying our model to estimate the age of infection. While the model can also predict whether an infection occurred by age one without serological testing, Fig. 2D indicates this approach may be less accurate.

These study designs inspired by our model have important implications for epidemiological research, as they enable, for the first time, accurate identification of the timing of RSV infection in infancy without the need for costly surveillance—a key step toward understanding how and when early-life RSV infection may cause asthma and other long-term respiratory diseases[3,6,9]. This remains relevant even in the era of RSV prevention products, as current RSV vaccines and monoclonal antibodies appear to prevent severe disease but not infection itself[10–12]. Beyond research, our model may also support public health efforts, helping to identify infants likely to be infected within the age window when risk of developing asthma and other long-term respiratory morbidity is highest[9].

Since there are few RSV surveillance studies, we lastly tested whether we could use bronchiolitis healthcare encounters from the PRIMA cohort to estimate model parameters. This required making two concessions. First, we needed a set of assumptions, the most important being that the weight functions for RSV and the set of pathogens that also cause bronchiolitis and whose seasonality matched RSV's were equal up to a constant of proportionality (see "Methods"). For example, if a six-month-old's risk of infection with RSV was twice as large as a one-month-old's, their risk of infection with the aforementioned pathogens was also twice as large. Second, we needed to estimate the nuisance function $c(a)$ representing the probability an infection at age $a$ turns into bronchiolitis requiring a healthcare encounter, which led to larger standard errors in the estimated weight function. Despite these concessions, the estimated weight function closely matched that from INSPIRE, and we used it to recapitulate the observed distribution of infection ages in INSPIRE. These results suggest that while additional assumptions and a substantially larger sample size are needed to utilize bronchiolitis data, they can be used to estimate model parameters if surveillance data are limited or not available.

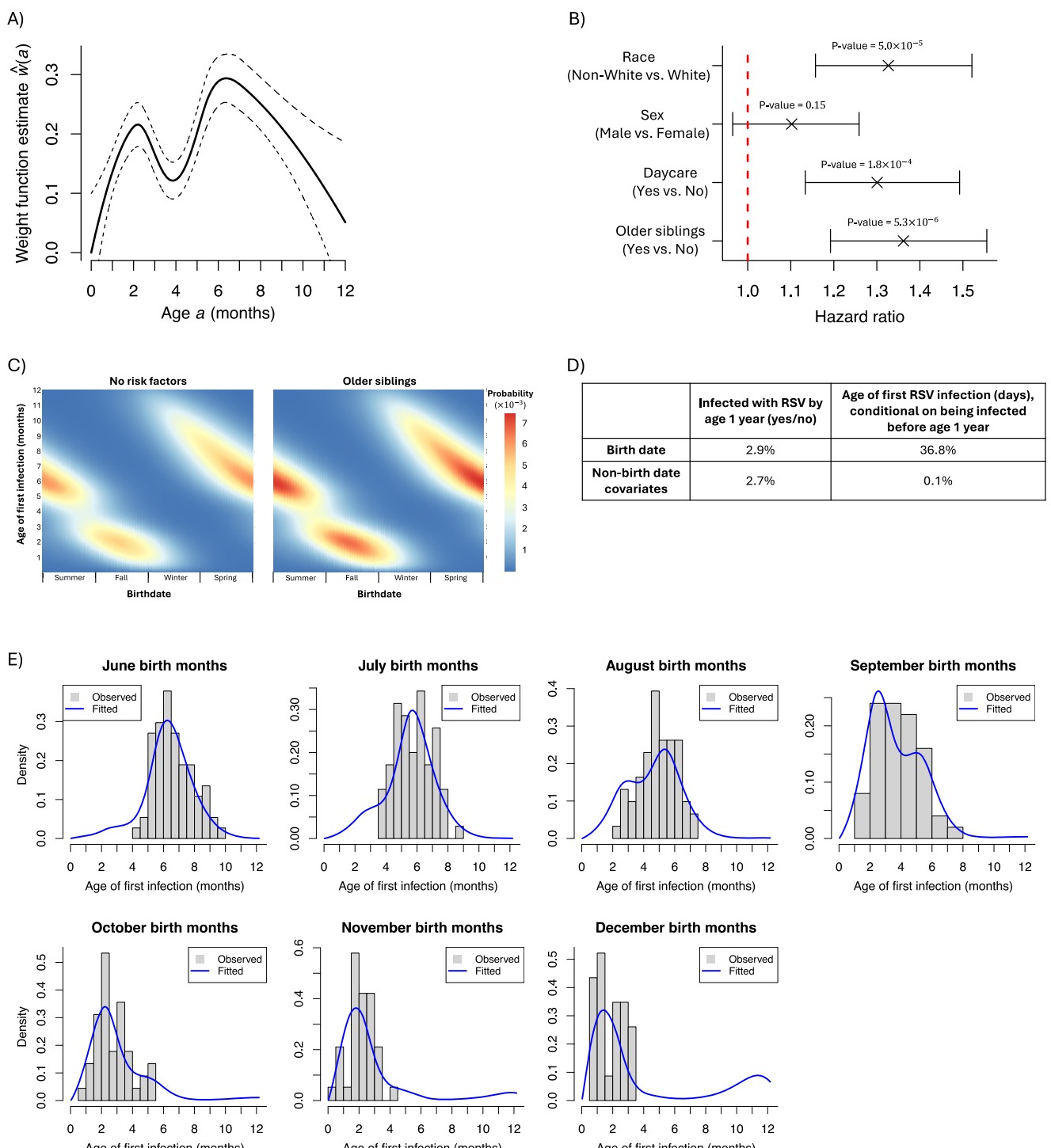

**Fig. 2 | Estimates from INSPIRE. A** The penalized maximum likelihood estimate for the weight function (solid line) and corresponding 95% confidence intervals (dashed lines; $n = 1713$). **B** Penalized maximum likelihood estimates for the effect of non-birthdate covariates and corresponding 95% confidence intervals ($n = 1713$). Estimates were exponentiated to get hazard ratios. *P*-values test the null hypothesis that the hazard ratio is 1 against the alternative that it is not equal to 1. They are not adjusted for multiple testing. **C** Heatmaps giving the estimated probability of first infection age in days as a function of birthdate for infants born in the geographical region covering INSPIRE participants from 2012-2013. Probabilities were estimated assuming no non-birthdate risk factors (left) and older siblings, but no other risk factors (right). **D** The percent of the variance of each outcome (columns) explained by each covariate (rows). **E** The average estimated probability density function for INSPIRE participants born in June through December. Probability densities were computed conditional on being infected before age one year.

The weight function estimated in the INSPIRE and PRIMA cohorts in Fig. 4A quantifies the age-dependent risk of RSV infection at a fixed point in time. It is close to zero for newborn infants, likely because newborns are both generally less exposed to the environment and may

be protected from RSV by maternal antibodies[23]. The peak at 6.5 months may be attributed to greater exposure to RSV at this age, and while maternal antibodies are not thought to prevent infection (they provide protection against severe disease), this period also coincides

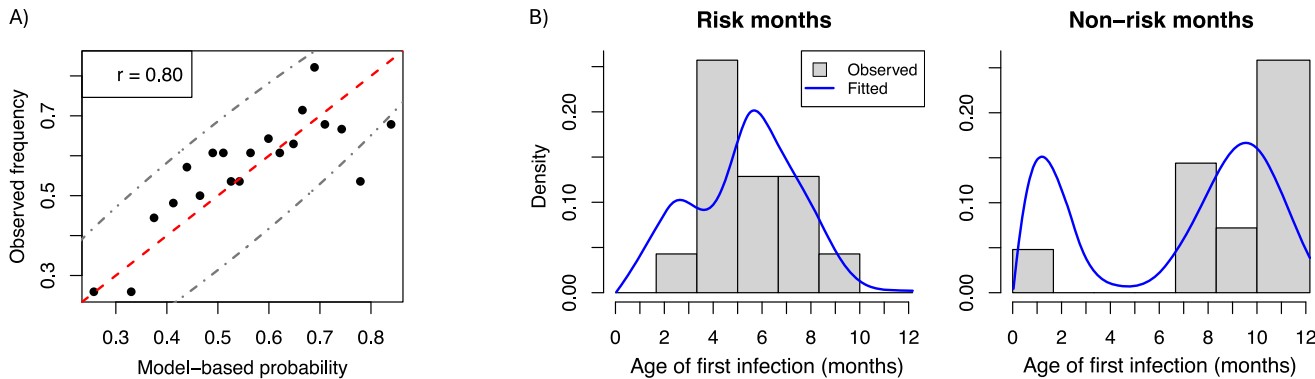

**Fig. 3 | Predicting age of first RSV infection in COAST and URECA using parameter estimates obtained from INSPIRE. A** Model calibration curve. COAST and URECA individuals were binned by their model-based probability they were infected by age one year; bins contained between 27 and 28 individuals. The x-axis is the average probability in each bin and the y-axis fraction of individuals in each bin that were actually infected by age one year. The dashed red line is the line $y = x$, dot-dashed gray lines are 95% confidence intervals, and $r$ is the correlation between $x$ and $y$ values. **B)** The estimated probability density function for COAST participants with a measured age of first infection before age one year that were born in risk months ($n = 28$) and non-risk months ($n = 25$). Fitted densities are conditional on being infected before age one year.

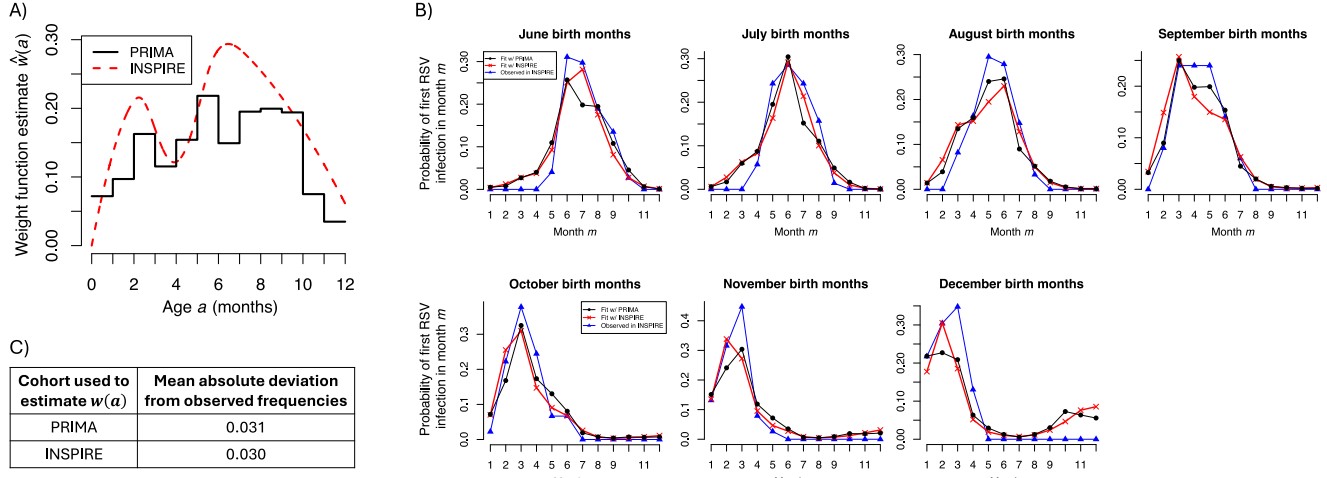

**Fig. 4 | Estimates derived from PRIMA compared to those from INSPIRE. A** Maximum likelihood estimate for the weight function. **B** The estimated probability individuals from INSPIRE were first infected with RSV in each month during the first year of life using the weight function estimated from PRIMA (black) or INSPIRE (red). The blue curves give the frequencies observed in INSPIRE. Probabilities are conditional on being infected in the first year of life. **C** The mean absolute deviation between the estimated probabilities and observed frequencies in (**B**).

with the waning of maternal RSV antibodies[23,24]. The estimate in INSPIRE also exhibits a clear bimodal pattern, which was present even after re-estimating parameters in different subpopulations of INSPIRE. Bimodality is also evident in the estimate from PRIMA, which, taken together with the generalizability of estimates from INSPIRE to other populations, suggests the pattern is not a statistical anomaly or an artifact of cohort design. The decrease from the first mode at two months to the nadir at four months could reflect an improved T cell-mediated immune response, as this coincides with the time that naïve CD4[+] T cells are highest in infants[25], which confer robust protection from lung infections in infancy[26].

Despite our model's strong performance, several limitations warrant consideration. First, due to the availability of surveillance data in INSPIRE, we modeled the age of first RSV infection only up to one year. Extending beyond this age is possible, but would require careful data curation to avoid misclassifying second-year infections as first infections. We avoided this issue since true repeat infections before age one are rare and typically represent unresolved primary infections[27]. Moreover, infections in the first year likely have the

greatest relevance for asthma-related outcomes[3,9]. Second, our cohorts predate the introduction of RSV prevention products. While this does not preclude using the model to investigate historical links between infection timing and asthma, whether our assumptions apply to infants who mothers received the RSV vaccine or infants who received a monoclonal antibody could not be determined. However, while vaccines and monoclonal antibodies effectively reduce disease severity, there is no evidence suggesting they prevent infection itself[10,11,28]. Supporting this, a recent study found that postfusion RSV F protein levels in infants given Nirsevimab were similar to those in placebo recipients[12], suggesting infection and an antibody response to natural infection still occurred—implying our model likely remains relevant for vaccinated infants, though further validation in vaccinated cohorts is certainly warranted. Lastly, INSPIRE did not include fully asymptomatic infections detected solely by surveillance, raising concerns that our model may reflect symptomatic infections. However, this is unlikely for three reasons: (i) minimal symptoms were required for nasal sampling in INSPIRE[3], (ii) our model generalized well to two external cohorts, including asymptomatic infections, and (iii)

estimates from INSPIRE closely matched those from PRIMA, where symptom status was explicitly modeled.

In summary, we have developed a probability model to predict the age of first RSV infection. Our model utilizes publicly available RSV circulation data from the CDC and an individual's birthdate, as well as other demographic covariates, to provide personalized RSV infection risk prediction. Our model is interpretable, fits the data well, generalizes across populations, and can even be fit using easy-to-obtain bronchiolitis healthcare encounter data. Our work represents an important development in viral surveillance, as it allows identification of the age of infant RSV infection within large populations without the need for active surveillance.

## Methods

Informed consent was obtained from the parents of all research participants. Studies in INSPIRE, URECA, and COAST were approved by the Institutional Review Board of Vanderbilt University Medical Center (IRB # 111299), the University of Wisconsin and Western Institutional Review Board (IRB # 20142570), and the Human Subjects Committee at the University of Wisconsin-Madison (IRB # H-2007-0044). Studies in PRIMA were approved by the Kaiser Permanente Northern California and Vanderbilt University Institutional Review Boards.

### Study design

We included data from four independent cohorts: Infant Susceptibility to Pulmonary Infections and Asthma Following RSV Exposure (INSPIRE), Childhood Origins of Asthma (COAST), Urban Environment and Childhood Asthma (URECA), and Prevention of RSV: Impact on Morbidity and Asthma (PRIMA). In our primary analysis, we used RSV surveillance data available in INSPIRE to estimate model parameters and tested the performance of our estimated model in COAST and URECA. We used bronchiolitis healthcare visits available in the PRIMA cohort to determine whether model parameters could also be estimated from easy-to-obtain healthcare visit data.

### Age of first RSV infection measured in the INSPIRE, COAST, and URECA cohorts

The age of first RSV infection was measured for individuals participating in the INSPIRE study, our discovery cohort, during their first year of life. Participants were born between June and December[19]. Briefly in INSPIRE, active and passive surveillance was conducted during infancy, with every two-week assessment of respiratory symptoms, and a nasal sample run for RSV PCR for those with symptoms. Since not all RSV infections manifest symptoms, all infants had RSV serology measured at age one year to determine which infants were infected by age one year[3].

Age of first infection in the COAST cohort, our validation cohort, was measured similarly[17,29,30]. Individuals missing an infection age were assigned an infection status (infected or not infected) using the same serological testing as in INSPIRE. In the URECA cohort, only one year of age RSV serology data were available.

### Bronchiolitis healthcare visits in the PRIMA study

We used bronchiolitis healthcare visits from infants ≤ 1 year of age that were a part of the PRIMA study to determine whether we could estimate model parameters from easy-to-obtain healthcare visits. We considered PRIMA individuals that were enrolled in the Tennessee Medicaid Program and were born prior to 2004, so that an asthma diagnosis at age six years would be available[31]. Healthcare visits were hospitalizations, emergency room visits, or physician office visits. Infants' age at the first bronchiolitis healthcare visit was determined.

### Determining the amount of circulating RSV over time

We used publicly available data from the CDC to determine $\lambda(t)$, the amount of circulating RSV over time $t$, for 10 different geographical regions in the United States defined by Health and Human Services (HHS) regional offices (1–10). Note $\lambda(t)$ is a continuous function of $t$. While $\lambda(t)$ is region-specific, we suppress its dependence on region to avoid excess notation. Since the below model only depends on knowing $\lambda(t)$ up to a constant of proportionality, we define $\lambda(t)$ here up to a multiplicative constant.

The data for each geographical region consisted of the number of viral samples and the fraction of them that were RSV positive, determined by culture, antigen, or PCR testing. Data were reported on a weekly basis starting in January 1995. We determined the fraction of positive tests over time using antigen testing because antigen tests were reported in the greatest number over the course of our study; PCR test results were not available prior to 2004 (Supplementary Fig. S5). We assumed that the amount of circulating RSV was proportional to the fraction of positive tests, and therefore defined $\lambda(t)$ to be the fraction of positive tests after smoothing and standardizing so that the maximum height of each peak was one (Fig. 1; see Section S2.1 for details). Standardizing accounted for the year-to-year variation in the frequency of RSV testing[32].

We mapped study locations in each cohort (Table 1) to geographical regions by determining the HHS office and Census Bureau division that covered them.

### Probability model for age of first RSV infection

Here we describe our model relating an individual's birthdate to their age of first RSV infection. We show how to incorporate additional covariates below. Let $i$ index an individual, $B_i$ be their birthdate, and $R_i$ be their age of first RSV infection. We modeled the age of first infection by modeling the instantaneous risk of first infection, which is defined as the likelihood an individual is infected at age $a$ given they have not been infected up until that point. This is also known as the hazard function in survival analysis.

Recall $\lambda(t)$ is proportional to the abundance of RSV at time $t$. Since infection risk ought to be proportional to the amount of RSV in the population, a naïve model would be to assume that instantaneous risk is proportional to RSV abundance:

$$\mathrm{pr}(R_i = a | R_i \geq a, B_i) \propto \lambda(a + B_i),$$

where $a + B_i$ is the time at which individual $i$ is age $a$. However, this model ignores the fact that one's infection risk also depends on age, as very young infants are likely less exposed to RSV (and the environment in general) and have more maternally derived RSV antibodies than older infants[1]. We therefore modified the above model to assume instantaneous risk depends on both RSV abundance and age:

$$\mathrm{pr}(R_i = a | R_i \geq a, B_i) = \lambda(a + B_i) w(a). \tag{1}$$

Here, $w(a)$ is a non-negative weight function that captures the age-dependent risk of being infected by RSV. Since very young infants have the least exposure to RSV and the greatest levels of maternally derived RSV antibodies, and exposure increases and maternally derived antibodies decrease over time[1], we expect $w(a)$ to be small when age $a$ is close to zero and get larger as $a$ increases.

The instantaneous risk model in Equation (1) completely determines the probability model relating age of first infection and birthdate, as it implies the likelihood for individual $i$ can be expressed as

$$\mathrm{pr}(R_i = a | B_i) = \lambda(a + B_i) w(a) \exp\left\{ -\int_0^a \lambda(x + B_i) w(x) dx \right\}. \tag{2}$$

### Estimating the weight function

Since the function $\lambda(\cdot)$ is known, the only unknown in the likelihood given by Equation (2) is the age-dependent weight function $w(a)$. We

assume $w(a)$ is a continuous and smooth function and require $w(a) \geq 0$ to ensure the likelihood is non-negative. As the functional form of $w(a)$ is unknown, we parameterize $w(a)$ using a cubic B-spline basis[33], which can approximate any smooth function arbitrarily well with a sufficient number of knots[34]. Since the B-spline basis elements are themselves non-negative, we encode the non-negativity constraint in $w(a)$ by requiring basis coefficients be non-negative.

We fit $w(a)$ using data from the INSPIRE cohort, which contains data for individuals with an observed age of first infection $R_i$ between zero and one year ($n = 361$), individuals known to be infected before age one year but without an observed $R_i$ ($n = 583$), and individuals known to have their first infection after age one year but without an observed $R_i$ ($n = 797$). Our estimator for $w(a)$ is a penalized maximum likelihood estimator, where the penalty encourages $w(a)$ to be smooth (i.e., a small second derivative). We set the number of knots in our B-spline parametrization of $w(a)$ to be 18 (10 internal knots and eight boundary knots), which provided a sufficiently flexible model. While this ostensibly begets a large number of model parameters, the penalty meant the effective number of parameters used to model $w(a)$ was 6.6[21], implying over-fitting was not an issue. Using a smaller number of knots resulted in a nearly identical estimate (Supplementary Fig. S6). Sections S2.2–S2.3 provide additional details.

#### Incorporating additional covariates into the model

Since our model from Equation (1) is a model on the instantaneous risk, otherwise known as the hazard function, we incorporate non-birthdate covariates using a proportional hazards assumption. That is, if $\mathbf{z}_i$ is a vector of non-birthdate covariates for individual $i$, we modify the instantaneous risk from Equation (1) to be

$$\text{pr}(R_i = a | R_i \geq a, B_i, \mathbf{z}_i) = \lambda(a + B_i) w(a) \exp(\mathbf{z}_i^\top \mathbf{g}), \quad (3)$$

implying the likelihood from Equation (2) becomes

$$\text{pr}(R_i = a | B_i, \mathbf{z}_i) = \lambda(a + B_i) w(a) \exp(\mathbf{z}_i^\top \mathbf{g}) \exp\left\{ -\exp(\mathbf{z}_i^\top \mathbf{g}) \int_0^a \lambda(x + B_i) w(x) dx \right\}. \quad (4)$$

We detail how we jointly estimate $w(a)$ and $\mathbf{g}$ in Section S2.3. We included the covariates sex (male/female), race (White/non-White), daycare attendance (yes/no), and older siblings (yes/no) in our analysis of INSPIRE data. We did not include early-life nutrition (e.g., breast milk or not) because much of an infant's protection to RSV derives from transplacental transfer of maternal antibodies[23,35], and because over 80% of INSPIRE individuals were breastfed.

#### Testing the estimated model in the COAST and URECA cohorts

We used our INSPIRE-derived estimates for $w(a)$ and $\mathbf{g}$ in Equation (4) to estimate, for each COAST and URECA participant, (i) the probability they were infected with RSV by age one year and (ii) their probability density (Equation (4)), which gives the likelihood they were first infected at age $a$ for all $a$ from 0 to 1 year. We included the same non-birthdate covariates as we did when fitting the model in INSPIRE (see above). Section S2.4 contains additional details.

#### Estimating parameters using bronchiolitis healthcare visit data

We used the bronchiolitis healthcare visit data from the PRIMA cohort to determine whether we could use easy-to-obtain healthcare encounter data to estimate the weight function $w(a)$. We ignored additional demographic covariates because we found their impact in INSPIRE to be minor (see Results).

It was not known whether bronchiolitis events in PRIMA were caused by RSV. While the seasonal abundance of RSV and frequency of bronchiolitis health care visits are congruent[9], and most bronchiolitis health care visits during infancy are caused by RSV[36-38], it is possible that other pathogens whose seasonality overlaps with RSV circulation could cause bronchiolitis. To address this, we assumed that the age-dependent variation in risk of being infected by these pathogens was the same as it was for RSV. That is, the weight functions for these pathogens and RSV were the same up to a constant of proportionality. For example, if a six-month-old was twice as likely to be infected by RSV as a one-month-old (i.e., $w(6 \text{ months}) = 2w(1 \text{ month})$), the risk of infection by these pathogens is also twice as large.

We additionally assumed that (i) each individual can be infected with RSV or the above mentioned pathogens at most once during the first year of life and (ii) the probability an infection turns into bronchiolitis requiring a healthcare visit depends on the infant's age and not the date of the infection. Assumption (i) is motivated by the observation that so-called "repeat RSV infections" during the first year of life are rare and, when they do occur, have been shown to represent the same virus, presumably because it has not been cleared by the host[27]. We assume (ii) because we know of no data suggesting otherwise.

Let $L_i$ be the age of the $i$-th individual's first bronchiolitis healthcare visit and $\tilde{R}_i$ be the age of their first RSV infection or their first infection with one of the above mentioned pathogens whose seasonality overlaps with RSV circulation. Under the above assumptions, the likelihood for $L_i$ conditional on birthdate $B_i$ is

$$\begin{aligned}
\text{pr}(L_i = a | B_i) &= \mathbb{P}(L_i = a | \tilde{R}_i = a) \text{pr}(\tilde{R}_i = a | B_i) \\
&= c(a) \lambda(a + B_i) \tilde{w}(a) \exp\left\{ -\int_0^a \lambda(x + B_i) \tilde{w}(x) dx \right\},
\end{aligned} \quad (5)$$

where $\tilde{w}(a)$ is proportional to $w(a)$ and $c(a)$ is unknown and is the probability an infection at age $a$ turns into bronchiolitis. As we did in INSPIRE, we estimated $\tilde{w}(a)$, as well as $c(a)$, up to age one year. Since only the years and months of birth and bronchiolitis healthcare visit dates were available in PRIMA, we assumed $\tilde{w}(a)$ and $c(a)$ were step functions with steps (i.e., discontinuities) at months one through 11. Since $\tilde{w}(a)$ is proportional to $w(a)$, we set our estimate for $w(a)$ to be a normalizing constant times our estimate for $\tilde{w}(a)$, where the normalizing constant was chosen so that the resulting model-based probability an infant was infected with RSV before age one year matched the frequency observed in INSPIRE. Section S2.5 contains additional details.

We used our PRIMA-derived estimate for $w(a)$ to estimate the probability INSPIRE individuals were first infected with RSV in each of their first 12 months of life. The mathematical expression is given in Section S2.6. We consider this probability and not the density like we did for COAST and URECA individuals because $w(a)$ could only be estimated at the resolution of months in PRIMA.

#### Reporting summary

Further information on research design is available in the Nature Portfolio Reporting Summary linked to this article.

### Data availability

RSV circulation data used in this study were obtained from the National Respiratory and Enteric Virus Surveillance System (NREVSS). The data can be requested from NREVSS at nrevss@cdc.gov. The raw data from the INSPIRE cohort used to estimate model parameters, as well as the raw data from the COAST and URECA cohorts used to test the estimated model, contain identifiable information such as birth dates and therefore cannot be made publicly available. Investigators may submit an asthma- or allergy-related application to use these data (https://cadre.med.wisc.edu). The process generally takes 1-2 months. Data from the PRIMA cohort cannot be shared as per our contract and data use agreement with TennCare. Individuals may apply to use TennCare data for research purposes (https://www.tn.gov/tenncare/legal/use-of-tenncare-data-for-research.html). The process usually

includes both TennCare and the Tennessee Department of Health approvals and may take months and depends on data availability and project scope.

## Code availability

All code, as well as instructions on how to run it, has been uploaded to https://github.com/chrismckennan/RSV-Infection-Age. Since we cannot share birth dates, we have provided tools to simulate birth dates and RSV infection ages, which can then be used to estimate model parameters.

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

## Acknowledgements

We are deeply grateful to all of the families and children who participated in this study, to the INSPIRE research study staff, to the middle Tennessee pediatric practices with whom we collaborated to enroll a representative population of our region, to the Division of TennCare in the Tennessee Department of Finance and Administration, and the Tennessee Department of Health, Office of Policy, Planning & Assessment. This work was supported by the National Institutes of Health grants U19AI095227 (C.G.M., T.G., T.V.H.), UG3OD023282 (T.G., T.V.H.), UL1TR000445 (T.G., P.W., T.V.H.), 1U24IO769079 (T.V.H.), R01HL173480 (C.G.M., P.W., T.V.H.) and the Agency for Healthcare Research and Quality grant R01HS018454 (T.V.H.). The content is solely the responsibility of the authors and does not necessarily represent the official views of the funding agencies.

## Author contributions

Conceptualization: C.G.M., T.G., P.W., and T.V.H. Methodology: C.G.M., T.G., P.W., and T.V.H. Investigation: C.G.M., T.G., S.M.B., P.W., and T.V.H. Visualizations: C.G.M. and T.V.H. Funding acquisition: D.J.J., J.E.G., P.W., and T.V.H. Writing: C.G.M. Writing – review & editing: C.G.M., T.G., P.W., S.M.B., M.N., D.J.J., J.E.G., P.W., and T.V.H.

## Competing interests

CGM reports grants from NIH during the conduct of the study and personal fees from SignatureDx outside the submitted work. TVH is a member of the NIH/NHLBI Council, the Parker B. Francis Foundation Council of scientific advisors as a grant reviewer, serves as the co-chair of the ATS Vaccines and Immunization Initiative, content writer for UpToDate, and a member of the RSV vaccine program DSMB for Pfizer. JEG is a consultant and has stock options in Meissa Vaccines Inc. The remaining authors declare no competing interests.
