## [Transparent Peer Review file · Nature Communications]

Predicting age of respiratory syncytial virus infection from birth timing

Corresponding Author: Professor Chris McKennan

Version 0:

Reviewer comments:

Reviewer #1

(Remarks to the Author)

Thank you for the opportunity to review this interesting manuscript. I have no major comments on the analysis itself. The approach seems reasonable and the modelling appears to be well done, particularly in the use of 3 different cohorts to develop and validate the model. My main comments relate to the practical utility of this work, which was not immediately apparent to me and didn't come out very strongly in the paper. I have a few comments in this regard below:

1. The analysis was based on data from historical cohorts. These are useful for model development, but are quite dated now, particularly since these all pre-date the introduction of RSV immunization for pregnant individuals and neonates. I would question the model's ability to accurately predict timing of first infection in an immunization setting, but this isn't really addressed in the paper.
2. It's not immediately obvious why estimating timing of first infection is a useful thing to do from a public health perspective. I can see that it might be useful to inform vaccination strategies, although I also see that these can and have been developed without this specific information, and I don't think vaccination recommendations would substantively change on the basis of this type of model. Similarly, we don't have good models of timing of first influenza infection, but I don't think these are specifically something that is missing from the public health response. Perhaps there are other uses for this analysis, such as to inform mathematical models of RSV transmission. Or maybe I'm missing something, but in any case it would be useful to make this clearer in the paper
3. The authors suggest that this approach could reduce surveillance burdens. As an example, they suggest that this model could be used in combination with serological surveillance at 12 months to determine age at first infection. I have two comments here: 1. surveillance activities should be tied to specific public health goals. I would argue that the purpose of RSV surveillance should not be to estimate incidence of infection, but rather to monitor trends in disease, particularly severe disease, assess the impact of immunization and, potentially, measure health system impacts of seasonal surges in demand. 2. It's not clear to me that serological surveillance for RSV in infants is a desirable or justifiable or acceptable thing to do, given that it's a somewhat invasive and/or intrusive, even if low risk, procedure that has no clear benefit to the child.

(Remarks on code availability)

Reviewer #2

(Remarks to the Author)

In this study, the authors develop an interesting probabilistic framework to infer the age of the first RSV infection in children. The survival model incorporates information on birth month, RSV activity, and demographic variables. The analysis is sound and described in detail. As noted, making analysis code available would make it easier to assess the specifics of the method. In general, this is a potentially useful tool and interesting and novel approach.

-An alternative or additional visualization might be to create a heatmap with birth month on one axis and age on the other axis, with the coloring reflecting the probability. This could allow for efficient comparisons across the studies and an

understanding of how covariates shift the highest risk periods.

-seasonal RSV incidence curves differ markedly by age, with nearly a month between the peak in infants and older adults. It is possible that the NREVSS data could be dominated by samples from older individuals, especially in more recent years just due to the sizes of the populations. How would this type of discrepancy affect the projections?

-Methods: "That is, $\lambda(t)$ actually gives the relative prevalence of RSV, where the relative amount of circulating RSV in a geographical region at any time t_1 and t_2 is $\lambda(t_1)/\lambda(t_2)$." It is not clear what this means or what was done here—why would the RSV detections at t_1 be divided by the RSV detections at t_2 ? Shouldn't the CDC data just be scaled to a specific range?

-To what extent can w_a be generalized between populations? The covariates will influence the hazard and provide some ability to have different hazard functions for different populations, but not as flexibly as w_a itself.

-Were any alternative structures for w_a considered, and how sensitive are the results to the chosen structure?

Methods: The formulae look technically correct, but they could likely be summarized more clearly for a general audience. For example, is $\lambda(\cdot)$ simply a function of RSV detection in NREVSS and a scalar? If so, can it be written as such? E.g.: $\Pr() = \alpha * \text{RSV_detections}_t * f(w_a) * \exp(X * B)$

-The data understandably cannot be shared. However, the code should be made available on a public repository such as Github, ideally with synthetic data that would allow for evaluation of the model properties.

(Remarks on code availability)

A link to the code is not included in the manuscript, and they say it is only available on request. I have included a comment on this in my review.

Version 1:

Reviewer comments:

Reviewer #1

(Remarks to the Author)

I appreciate the opportunity to review this revised manuscript and thank the authors for their thoughtful responses. The additional context makes the rationale and utility of the paper much clearer, and my original comments have been adequately addressed. I have no additional comments.

(Remarks on code availability)

I have not reviewed the code in detail, though I note that the files provided include model parameters and scripts to derive the main analysis, as well as a worked example in R markdown.

Reviewer #2

(Remarks to the Author)

Overall the authors have done a thorough job in responding to my suggestions.

As a minor point of clarification: the authors rebutted my suggestion "Seasonal RSV incidence curves differ markedly by age, with nearly a month between the peak in infants and older adults. It is possible that the NREVSS data could be dominated by samples from older individuals, especially in more recent years just due to the sizes of the populations. How would this type of discrepancy affect the projections?"

they rebut this point by saying "as can be seen in Figure R2 in this document, which give RSV hospitalization rates in infants and adults obtained from <https://www.cdc.gov/rsv/php/surveillance/rsv-net.html>, there is very little difference between incidence across these age groups. Notably, peak incidence differs by only 1 week in the 2018-2019 season and is the same in the 2023-2024 and 2024-2025 seasons. There is therefore little to no evidence suggesting that age has an impact on RSV infection incidence estimates"

It appears this curve is made using summary estimates across the different sites. If the authors repeat this analysis for individual sites, they will find a 3-4 week gap between the peak in incidence for many sites and many years. Similar gaps are found in emergency department data from across the US, as summarized here: <https://www.pophive.org/respiratory-diseases/respiratory-synycial-virus>

(Remarks on code availability)

Response to Reviewers

We are grateful for the thoughtful comments and constructive advice from the two reviewers, as their comments have substantially improved our manuscript. We include all comments in slanted text and respond to each concern in regular text. Please note that figures, tables, and equations that appear in this review are in the form “R[0-9]+”.

For the Reviewers’ convenience, we have highlight below the parts of the manuscript that have changed the most.

Parts of the manuscript that have changed the most

Introduction

The first and second paragraphs of Introduction have been modified to address concerns raised by Reviewer 1.

- **Implications of our work.** We now explicitly state the implications and importance of our work in the second paragraph of Introduction. Specifically, we note our study’s importance in addressing critical epidemiological and basic science questions regarding the causal role of RSV infection in long-term respiratory outcomes, immune and airway development, and chronic disease. We also discuss its potential utility in public health.
- **The relevance of our findings in the era of RSV prevention products.** This is now addressed in the last sentence of the first paragraph, where we note that the current literature suggests that while RSV vaccines and monoclonal antibodies reduce the severity of RSV infection, they do not prevent infection itself.

Discussion

We have modified the fourth, fifth, and penultimate paragraphs of Discussion to address concerns raised by Reviewer 1.

- **Fourth and fifth paragraphs.** In the fourth paragraph we discuss how our results imply we can identify infection age without having to perform time- and cost-intensive RSV surveillance. The fifth paragraph details, as we did above in response to Reviewer 1’s second comment, why this is of critical importance to epidemiological research studying the impact of infection timing on childhood asthma risk and other diseases. It also discusses why our work remains important in the era of RSV vaccines and monoclonal antibodies, and our model’s potential immediate public health impact.
- **Penultimate paragraph.** We have modified this paragraph by adding that a limitation of our study was that cohorts used in this study predate the introduction of RSV prevention products. However, as we discussed above, we noted that existing evidence suggests these do not prevent infection itself, indicating our model may still be relevant for vaccinated infants.

Reviewer 1

Thank you for the opportunity to review this interesting manuscript. I have no major comments on the analysis itself. The approach seems reasonable and the modeling appears to be well done, particularly in the use of 3 different cohorts to develop and validate the model. My main comments relate to the practical utility of this work, which was not immediately apparent to me and didn't come out very strongly in the paper. I have a few comments in this regard below:

- 1. The analysis was based on data from historical cohorts. These are useful for model development, but are quite dated now, particularly since these all pre-date the introduction of RSV immunization for pregnant individuals and neonates. I would question the model's ability to accurately predict timing of first infection in an immunization setting, but this isn't really addressed in the paper.*

We thank the Reviewer for their comment. The timing of a child's first RSV infection is influenced largely by environmental exposures and viral transmission dynamics. While it is true that models predicting the timing of first infection may have inherent limitations—especially when data predates the introduction of RSV prevention products—it is crucial to distinguish between preventing infection altogether and mitigating severe disease outcomes. RSV prevention products, such as monoclonal antibodies and vaccines, were designed to prevent severe manifestations of RSV rather than prevent infection itself [1, 2, 3]. This distinction underscores their role in public health strategies aimed at reducing hospitalization rates, ICU admissions, and mortality rather than altering broader transmission patterns. Notably, recent work showing that the RSV F protein was present at similar levels in the serum of placebo- and Nirsevimab-treated patients suggests that RSV vaccines do not prevent infections and immune responses [4]. It is therefore reasonable to hypothesize that the introduction of RSV prevention products might not significantly change the timing of first infection. However, we are in full agreement with the reviewer that future research assessing infection timing in the post-introduction of RSV prevention products era (and comparing with the pre-introduction timing) will be important.

Our revised manuscript now includes the following to incorporate the above discussion:

- **The first paragraph of Introduction:** The last sentence of the first paragraph acknowledges that vaccines have been developed to reduce disease severity, but also notes that the existing literature suggests they do not prevent infection or host immune response.
 - **Penultimate paragraph of Discussion:** We have modified this paragraph by adding that a limitation of our study was that cohorts used in this study predate the introduction of RSV vaccines. However, as we discussed above, we noted that existing evidence suggests RSV vaccines and RSV monoclonal antibodies do not prevent infection itself or host antibody response to infection, indicating our model may still be relevant for infants vaccinated through maternal vaccine or infant receipt of Nirsevimab.
- 2. It's not immediately obvious why estimating timing of first infection is a useful thing to do from a public health perspective. I can see that it might be useful to inform vaccination strategies, although I also see that these can and have been developed without this specific information, and I don't think vaccination recommendations would substantively change on the basis of this type of model. Similarly, we don't have good models of timing of first influenza infection, but I*

don't think these are specifically something that is missing from the public health response. Perhaps there are other uses for this analysis, such as to inform mathematical models of RSV transmission. Or maybe I'm missing something, but in any case it would be useful to make this clearer in the paper

We thank the Reviewer for their comment, and apologize for not elaborating on this further in the manuscript. Our group has previously shown that early-life (≤ 1 year) infection with RSV—defined using active and passive surveillance with qPCR- and serology-identified infection—is likely causal for childhood asthma and other respiratory sequelae [5], where the effect may be mediated by the impact of RSV on airway barrier development, metabolism, the microbiome, and epigenetics [6, 7, 8, 9, 10]. The model presented in this manuscript was therefore specifically motivated by key unanswered research questions regarding the impact of early-life RSV infection on respiratory health, immune system development, airway epithelial development, and susceptibility to future infections or chronic conditions, and whether there is a critical susceptibility period (i.e., age) when risk of RSV infection is associated with the highest risk of developing asthma [11, 5]. Such insights are critical for advancing scientific understanding of RSV pathology and informing the design of targeted interventions to prevent long-term respiratory morbidity. To address such questions, it is necessary to identify first infection age in large cohorts. Unfortunately, what has been historically studied—severe RSV disease requiring hospitalization—only occurs in approximately 2% of all infants and is therefore not representative of general infections [12, 13]. To complicate matters further, most infections are asymptomatic [5]. Consequently, addressing these research questions currently requires costly RSV surveillance that is prohibitively expensive and time consuming in large cohorts. Our work therefore represents a pivotal research advancement by allowing, for the first time, accurate estimation of first infection age without the need for costly surveillance.

As we noted in our response to the Reviewer's first comment, our research motivations and model are still of interest in the era of RSV prevention products, as vaccines and monoclonal antibodies do not appear to prevent infection or the host's immune response [1, 2, 4]. Unlike ongoing infection surveillance throughout RSV seasons, RSV sero-surveillance for research purposes offers a more efficient, cost-effective, and minimally burdensome approach for research participants. By focusing on serological markers, this method provides a robust tool for retrospective identification of infection timing without the logistical challenges inherent in active surveillance systems. The development of this model highlights the intersection of scientific innovation and practical feasibility, paving the way for future research studies that could ultimately inform both research priorities and potentially public health strategies.

Our model also offers an immediate potential public health benefit. As noted in our response to the Reviewer's first comment above, RSV vaccines and monoclonal antibodies appear to prevent severe clinical manifestations of infection but not infection itself [1, 2, 3, 4]. Given our previous findings that RSV infection is likely *causal* in the development of childhood wheeze and asthma [5], this highlights the value of our model in addressing the critical need to identify infants at risk of infection during the age window when susceptibility to asthma is highest.

We have now made the following changes to the manuscript to reflect the above discussion:

- **First and second paragraphs of Introduction:** In the first paragraph, we emphasize the importance of studying RSV infection itself—as distinct from severe infection, which accounts for only 2% of all infections. In the second we discuss why identifying infants that were, or will be, infected with RSV during the abovementioned critical susceptibility

age window is so important to both epidemiological research and public health.

- **Fourth and fifth paragraphs of Discussion:** In the fourth paragraph we discuss how our results imply we can identify infection age without having to perform costly RSV surveillance. The fifth paragraph details, as we did above in response to the Reviewer’s comment, why this is of critical importance to epidemiological research studying the impact of infection timing on childhood asthma risk. It also discusses why our work remains important in the era of RSV prevention products, and our model’s potential immediate public health impact.

3. *The authors suggest that this approach could reduce surveillance burdens. As an example, they suggest that this model could be used in combination with serological surveillance at 12 months to determine age at first infection. I have two comments here:*

- (a) *Surveillance activities should be tied to specific public health goals. I would argue that the purpose of RSV surveillance should not be to estimate incidence of infection, but rather to monitor trends in disease, particularly severe disease, assess the impact of immunization and, potentially, measure health system impacts of seasonal surges in demand.*

We thank the Reviewer and agree with the comment that surveillance should be tied to specific public health goals. We clarified and updated Introduction and Discussion that this model could reduce “research oriented” surveillance burdens (see our response to their first and second comments). For future research studies with an interest in age at RSV infection, instead of collecting data frequently, our model, by measuring infection status at age one year, provides a resource saving and low risk/low burden alternative.

- (b) *It’s not clear to me that serological surveillance for RSV in infants is a desirable or justifiable or acceptable thing to do, given that it’s a somewhat invasive and/or intrusive, even if low risk, procedure that has no clear benefit to the child.*

We thank the Reviewer for their comment. As we noted above in our response to part (a), serological surveillance would be done for research purposes.

Reviewer 2

In this study, the authors develop an interesting probabilistic framework to infer the age of the first RSV infection in children. The survival model incorporates information on birth month, RSV activity, and demographic variables. The analysis is sound and described in detail. As noted, making analysis code available would make it easier to assess the specifics of the method. In general, this is a potentially useful tool and interesting and novel approach.

- 1. An alternative or additional visualization might be to create a heatmap with birth month on one axis and age on the other axis, with the coloring reflecting the probability. This could allow for efficient comparisons across the studies and an understanding of how covariates shift the highest risk periods.*

We thank the Reviewer for their comment, and think this is a great idea. We have now added two heat maps to Figure 2 (see panel C in the revised manuscript), which is a multi-panel figure detailing estimates derived from the INSPIRE cohort. The figure has been included in this document for the Reviewer’s convenience (see Figure R1). We have also updated the end of the second paragraph of “Parameter estimates and model evaluation” in Results to discuss the implications, which we have included below for the Reviewer’s convenience:

“Having estimated parameters, we then investigated the relationship between birthdate and infection risk, as well as how the relationship changed with non-birthdate covariates. As indicated in Figure 2C, infants born in June (summer) are most likely to be infected at 6 months, whereas those born in December (late fall, early winter) are most likely to be first infected at 1 to 3 months. Non-birthdate covariates appear to only alter the magnitude, not the shape, of the relationship between birthdate and infection risk.”

- 2. Seasonal RSV incidence curves differ markedly by age, with nearly a month between the peak in infants and older adults. It is possible that the NREVSS data could be dominated by samples from older individuals, especially in more recent years just due to the sizes of the populations. How would this type of discrepancy affect the projections?*

We thank the Reviewer for their comment. We respectfully disagree with the Reviewer’s

Figure R1: Heatmaps giving the estimated probability of first infection as a function of birthdate for infants born in the geographical region covering INSPIRE participants from 2012-2013. Probabilities were estimated assuming no non-birthdate risk factors (left) and older siblings but no other risk factors (right).

Figure R2: RSV hospitalization rates in infants and older adults living in the United States. Data were obtained for non-COVID RSV seasons for which data in both age cohorts was available.

assertion that RSV incidence curves differ by age. As can be seen in Figure R2 in this document, which give RSV hospitalization rates in infants and adults obtained from <https://www.cdc.gov/rsv/php/surveillance/rsv-net.html>, there is very little difference between incidence across these age groups. Notably, peak incidence differs by only 1 week in the 2018-2019 season and is the same in the 2023-2024 and 2024-2025 seasons. There is therefore little to no evidence suggesting that age has an impact on RSV infection incidence estimates.

3. *Methods*: “That is, $\lambda(t)$ actually gives the relative prevalence of RSV, where the relative amount of circulating RSV in a geographical region at any time t_1 and t_2 is $\lambda(t_1)/\lambda(t_2)$.” It is not clear what this means or what was done here—why would the RSV detections at t_1 be divided by the rsv detections at t_2 ? Shouldn’t the CDC data just be scaled to a specific range?

We thank the Reviewer for their comment, and apologize for any confusion. To avoid any future confusion, we no longer use “RSV prevalence” in the revised manuscript when referring to $\lambda(t)$, as we do not mean for this to be taken to be the number of RSV cases at any time. We now refer to it as “RSV abundance”, since $\lambda(t)$ actually represents the amount of circulating RSV at time t . We have also removed the sentence the Reviewer references in their comment to avoid additional confusion.

Our implicit assumption is that $\lambda(t)$ is proportional to the fraction of positive tests at time t in the geographic region of interest. Since the constant of proportionality is unknown, we can only know $\lambda(t)$ up to a constant. However, that has no impact on our model, since our model only requires knowing $\lambda(t)$ up to a constant. We have now included the following sentence in the second paragraph of “Determining RSV prevalence over time” in Methods to make this clear:

“We assumed that the amount of circulating RSV was proportional to the fraction of positive tests, and therefore defined $\lambda(t)$ to be the fraction of positive tests after smoothing and standardizing so that the maximum height of each peak was one (Fig. 1; see Section S2.1 for details).”

4. *To what extent can w_a be generalized between populations? The covariates will influence the hazard and provide some ability to have different hazard functions for different populations, but not as flexibly as w_a itself.*

This is an excellent question, and one that we worked really hard to study. Specifically, we showed the following in Results:

- “Testing our estimated model in two independent cohorts”: here we showed that the model

trained in INSPIRE could be used to successfully predict RSV infection timing in two independent cohorts.

- “Using bronchiolitis healthcare visits to estimate model parameters”: here we estimated $w(a)$ using bronchiolitis healthcare visit data from the PRIMA cohort. As we showed in Figure 4 and discussed in the corresponding text, the estimates for $w(a)$ were remarkably similar, where the estimate derived from PRIMA could be used to predict infection timing in INSPIRE.

Taken together, these indicate $w(a)$ generalizes across races/ethnicities, geographic locations, study periods, and birth months. We have now added the following to the third paragraph of Discussion to explicitly note that our results suggest $w(a)$ generalizes across populations:

“As the two testing cohorts were starkly different from INSPIRE in terms of their racial composition, geographic locations, study periods, and birth months (Table 1), these results indicate that our proposed model, model parameters (e.g., the weight function), and parameter estimates generalize across populations.”

5. *Were any alternative structures for w_a considered, and how sensitive are the results to the chosen structure?*

We thank the Reviewer for the comment. We used a cubic B-spline basis to parametrize $w(a)$. Given a set of knots, the cubic B-spline basis spans all cubic splines with those knots, where cubic splines can approximate any twice-differentiable (C^2) function arbitrarily well with a sufficient number of knots [14]. (This is the reason why the R package `gam` uses splines as their default method of obtaining non-parametric function estimates [15].) Consequently, our approach can approximate any smooth function provided we choose a sufficient number of knots. As such, we can explore the robustness of our estimate by varying the number of knots. (Please note that as we described in Methods, we penalized the second derivative of $w(a)$ to avoid over-fitting when the number of knots is large.)

We used 18 knots (10 internal and 8 boundary knots) to estimate $w(a)$ in our manuscript, which is likely substantially more than needed, as the effective number of parameters in our model for $w(a)$ was only 6.6 (see Methods). We also showed that using 13 knots (5 internal and 8 boundary) resulted in a nearly identical estimate for $w(a)$ (see Figure S6 in the Supplement, which was referenced in Methods). Finally, we show in this document that 23 knots (15 internal and 8 boundary) gives a nearly identical estimate (Figure R3). Taken together, these results suggest 18 knots is more than sufficient to parametrize $w(a)$ and that our estimate is robust to the number of knots. We have now included the following text in “Estimating the weight function” in Methods to emphasize the B-splines are a natural choice for non-parametric function estimation because they can approximate any smooth function with an arbitrary number of knots:

“As the functional form of $w(a)$ is unknown, we parameterize $w(a)$ using a cubic B-spline basis [16], which can approximate any smooth function arbitrarily well with a sufficient number of knots [14].”

6. *Methods: The formulae look technically correct, but they could likely be summarized more clearly for a general audience. For example, is $\lambda(\cdot)$ simply a function of RSV detection in NREVSS and a scalar? If so, can it be written as such? E.g.: $Pr() = \alpha * RSV_detections_t * f(w_a) * \exp(X * B)$.*

Figure R3: Weight function estimates using 18 and 23 knots. Dashed lines give the 95% confidence intervals for the estimate with 18 knots.

We thank the Reviewer for their comment. We have chosen to keep the formulae as they were originally presented for several reasons. First, $\lambda(t)$ and $w(a)$ are continuous functions of time t and age a , so we feel it is appropriate to use functional notation (e.g., not subscripts). We have now made it clear that these are functions in Methods (see the second sentence of “Determining RSV prevalence over time” and the first sentence after Equation 1 in “Probability model for age of first RSV infection”). Second, while $\lambda(t)$ is derived from NREVSS after smoothing and standardizing their data, in our model it is interpretable as the amount of circulating RSV at time t , not RSV detection rate. Third, we unfortunately do not know what the Reviewer is referring to when they reference “alpha” and the function “ $f(\cdot)$ ” in their probability definition. Fourth, we feel that the formulae are composed of elementary symbols and expressions (e.g., “exp” and “ \int ”) that are well-known in the scientific community.

7. *The data understandably cannot be shared. However, the code should be made available on a public repository such as Github, ideally with synthetic data that would allow for evaluation of the model properties.*

We thank the Reviewer for their comment. We have now provided R scripts that allow one to simulate data and estimate the weight function and effect of non-birthdate covariates (see “CodeForReview.zip”). The README file contains instructions on how to run the code. This code will be uploaded to a public GitHub repository hosted by the corresponding author (<https://github.com/chrismckennan>) if the manuscript is accepted for publication.

8. *A link to the code is not included in the manuscript, and they say it is only available on request. I have included a comment on this in my review.*

Please see our response to the Reviewer’s comment above. The above code will be uploaded to a public GitHub repository hosted by the corresponding author (<https://github.com/chrismckennan>) if the manuscript is accepted for publication.

References

- [1] M. P. Griffin, Y. Yuan, T. Takas, J. B. Domachowske, S. A. Madhi, P. Manzoni, et al. “Single-Dose Nirsevimab for Prevention of RSV in Preterm Infants”. In: *New England Journal of Medicine* 383.5 (July 2020), pp. 415–425. ISSN: 1533-4406. DOI: 10.1056/nejmoa1913556. URL: <http://dx.doi.org/10.1056/NEJMoa1913556>.
- [2] L. L. Hammitt, R. Dagan, Y. Yuan, M. Baca Cots, M. Bosheva, S. A. Madhi, et al. “Nirsevimab for Prevention of RSV in Healthy Late-Preterm and Term Infants”. In: *New England Journal of Medicine* 386.9 (Mar. 2022), pp. 837–846. ISSN: 1533-4406. DOI: 10.1056/nejmoa2110275. URL: <http://dx.doi.org/10.1056/NEJMoa2110275>.
- [3] J. P. Torres, D. Sauré, M. Goic, C. Thraves, J. Pacheco, J. Burgos, et al. “Effectiveness and impact of Nirsevimab in Chile during the first season of a national immunisation strategy against RSV (NIRSE-CL): a retrospective observational study”. In: *The Lancet Infectious Diseases* (June 2025). ISSN: 1473-3099. DOI: 10.1016/s1473-3099(25)00233-6. URL: [http://dx.doi.org/10.1016/S1473-3099\(25\)00233-6](http://dx.doi.org/10.1016/S1473-3099(25)00233-6).
- [4] D. Wilkins, Y. Yuan, Y. Chang, A. A. Aksyuk, B. S. Núñez, U. Wählby-Hamrén, et al. “Durability of neutralizing RSV antibodies following nirsevimab administration and elicitation of the natural immune response to RSV infection in infants”. In: *Nature Medicine* 29.5 (Apr. 2023), pp. 1172–1179. ISSN: 1546-170X. DOI: 10.1038/s41591-023-02316-5. URL: <http://dx.doi.org/10.1038/s41591-023-02316-5>.
- [5] C. Rosas-Salazar, T. Chirkova, T. Gebretsadik, J. D. Chappell, R. S. Peebles, W. D. Dupont, et al. “Respiratory syncytial virus infection during infancy and asthma during childhood in the USA (INSPIRE): a population-based, prospective birth cohort study”. In: *The Lancet* 401.10389 (May 2023), pp. 1669–1680. DOI: 10.1016/s0140-6736(23)00811-5. URL: [https://doi.org/10.1016/s0140-6736\(23\)00811-5](https://doi.org/10.1016/s0140-6736(23)00811-5).
- [6] C. Rosas-Salazar, M. H. Shilts, A. Tovchigrechko, S. Schobel, J. D. Chappell, E. K. Larkin, et al. “Differences in the Nasopharyngeal Microbiome During Acute Respiratory Tract Infection With Human Rhinovirus and Respiratory Syncytial Virus in Infancy”. In: *Journal of Infectious Diseases* 214.12 (Dec. 2016), pp. 1924–1928. ISSN: 1537-6613. DOI: 10.1093/infdis/jiw456. URL: <http://dx.doi.org/10.1093/infdis/jiw456>.
- [7] A. R. Connelly, B. M. Jeong, M. E. Coden, J. Y. Cao, T. Chirkova, C. Rosas-Salazar, et al. “Metabolic Reprogramming of Nasal Airway Epithelial Cells Following Infant Respiratory Syncytial Virus Infection”. In: *Viruses* 13.10 (Oct. 2021), p. 2055. ISSN: 1999-4915. DOI: 10.3390/v13102055. URL: <http://dx.doi.org/10.3390/v13102055>.
- [8] C. McKennan, D. C. Newcomb, S. Berdnikovs, T. Gebretsadik, S. Ma, L. B. Bacharier, et al. “Impact of early-life respiratory syncytial virus infection on cell type-specific airway DNA methylation”. In: (Sept. 2024). DOI: 10.1101/2024.09.29.615688. URL: <http://dx.doi.org/10.1101/2024.09.29.615688>.
- [9] S. Berdnikovs, D. C. Newcomb, and T. V. Hartert. “How early life respiratory viral infections impact airway epithelial development and may lead to asthma”. In: *Frontiers in Pediatrics* 12 (Aug. 2024). ISSN: 2296-2360. DOI: 10.3389/fped.2024.1441293. URL: <http://dx.doi.org/10.3389/fped.2024.1441293>.

- [10] S. Berdnikovs, D. C. Newcomb, N.-F. Haruna, K. E. McKernan, S. N. Kuehnle, T. Gebretsadik, et al. “Single-cell profiling demonstrates the combined effect of wheeze phenotype and infant viral infection on airway epithelial development”. In: *Science Advances* 11.21 (May 2025). ISSN: 2375-2548. DOI: 10.1126/sciadv.adr9995. URL: <http://dx.doi.org/10.1126/sciadv.adr9995>.
- [11] P. Wu, W. D. Dupont, M. R. Griffin, K. N. Carroll, E. F. Mitchel, T. Gebretsadik, et al. “Evidence of a Causal Role of Winter Virus Infection during Infancy in Early Childhood Asthma”. In: *American Journal of Respiratory and Critical Care Medicine* 178.11 (Dec. 2008), pp. 1123–1129. ISSN: 1535-4970. DOI: 10.1164/rccm.200804-579oc. URL: <http://dx.doi.org/10.1164/rccm.200804-5790C>.
- [12] J. G. Wildenbeest, M.-N. Billard, R. P. Zuurbier, K. Korsten, A. C. Langedijk, P. M. van de Ven, et al. “The burden of respiratory syncytial virus in healthy term-born infants in Europe: a prospective birth cohort study”. In: *The Lancet Respiratory Medicine* 11.4 (Apr. 2023), pp. 341–353. DOI: 10.1016/s2213-2600(22)00414-3. URL: [https://doi.org/10.1016/s2213-2600\(22\)00414-3](https://doi.org/10.1016/s2213-2600(22)00414-3).
- [13] S. M. Brunwasser, B. M. Snyder, A. J. Driscoll, D. B. Fell, D. A. Savitz, D. R. Feikin, et al. “Assessing the strength of evidence for a causal effect of respiratory syncytial virus lower respiratory tract infections on subsequent wheezing illness: a systematic review and meta-analysis”. In: *The Lancet Respiratory Medicine* 8.8 (Aug. 2020), pp. 795–806. DOI: 10.1016/s2213-2600(20)30109-0. URL: [https://doi.org/10.1016/s2213-2600\(20\)30109-0](https://doi.org/10.1016/s2213-2600(20)30109-0).
- [14] C. A. Hall and W. Meyer. “Optimal error bounds for cubic spline interpolation”. In: *Journal of Approximation Theory* 16.2 (Feb. 1976), pp. 105–122. ISSN: 0021-9045. DOI: 10.1016/0021-9045(76)90040-x. URL: [http://dx.doi.org/10.1016/0021-9045\(76\)90040-X](http://dx.doi.org/10.1016/0021-9045(76)90040-X).
- [15] T. Hastie. *gam: Generalized Additive Models*. R package version 1.22-5. 2024. URL: <https://CRAN.R-project.org/package=gam>.
- [16] P. H. C. Eilers and B. D. Marx. “Flexible smoothing with B-splines and penalties”. In: *Statistical Science* 11.2 (May 1996). ISSN: 0883-4237. DOI: 10.1214/ss/1038425655. URL: <http://dx.doi.org/10.1214/ss/1038425655>.

Response to Reviewer 2

We are grateful for the thoughtful comments and constructive advice from the reviewer. We include all comments in slanted text and respond to each concern in regular text. Please note that figures, tables, and equations that appear in this review are in the form “R[0-9]+”.

1. *As a minor point of clarification: the authors rebutted my suggestion “Seasonal RSV incidence curves differ markedly by age, with nearly a month between the peak in infants and older adults. It is possible that the NREVSS data could be dominated by samples from older individuals, especially in more recent years just due to the sizes of the populations. How would this type of discrepancy affect the projections?”*

They rebut this point by saying “as can be seen in Figure R2 in this document, which give RSV hospitalization rates in infants and adults obtained from <https://www.cdc.gov/rsv/php/surveillance/rsv-net.html>, there is very little difference between incidence across these age groups. Notably, peak incidence differs by only 1 week in the 2018-2019 season and is the same in the 2023-2024 and 2024-2025 seasons. There is therefore little to no evidence suggesting that age has an impact on RSV infection incidence estimates.”

It appears this curve is made using summary estimates across the different sites. If the authors repeat this analysis for individual sites, they will find a 3-4 week gap between the peak in incidence for many sites and many years. Similar gaps are found in emergency department data from across the US, as summarized here: <https://www.pophive.org/respiratory-diseases/respiratory-syncytial-virus>.

We thank the Reviewer for their comment. We repeated the analysis and confirmed that, when considering individual sites, peak incidence occurred later in older adults (≥ 65 years) than in infants (≤ 1 year). The median delay across sites was 2 weeks, 1 week, and 3 weeks in the 2018-2019, 2023-2024, and 2024-2025 seasons, respectively.

Figure R1: RSV abundance estimates for the 2012-2013 RSV season in the geographic region covering INSPIRE participants before (black) and after (red) shifting by three weeks.

We first note that, given our results showing that the estimated model accurately predicts infection ages across races/ethnicities, geographic locations, study periods, and birth months, any errors in RSV abundance estimates—if present—are likely to have only a negligible impact on infection age predictions. To assess this directly, we used our estimated model to compare predicted infection ages for INSPIRE participants before and after shifting the RSV abundance curves three weeks to the right (see Figure R1 in this document). Consistent with the robustness mentioned above, the predicted infection ages changed very little. The average relative change

in the predicted probability of infection by age one year was 5%. For instance, a predicted probability of 0.70 before shifting increased to approximately 0.735 after shifting. Additionally, the expected infection age conditional on infection before age one year increased by an average of 9.3 days (standard deviation = 12.1 days). Taken together, these results suggest model-based predictions are robust to errors in estimates for RSV abundance.

We have now added these results to section S1.2 of the revised Supplement.